# The Placenta–Gut Microbiota Axis in Gestational Diabetes Mellitus: Molecular Mechanisms, Crosstalk, and Therapeutic Perspectives

**DOI:** 10.3390/ijms27010312

**Published:** 2025-12-27

**Authors:** Reka Anna Vass, Eva Miko, Viktoria Premusz, Sandor G. Vari, Kalman Kovacs, Jozsef Bodis, Tibor Ertl

**Affiliations:** 1Department of Obstetrics and Gynecology, Medical School, Clinical Centre, University of Pécs, H-7624 Pécs, Hungary; kovacs.kalman@pte.hu (K.K.); bodis.jozsef@pte.hu (J.B.); tibor.ertl@aok.pte.hu (T.E.); 2National Laboratory on Human Reproduction, University of Pécs, H-7622 Pécs, Hungary; miko.eva@pte.hu (E.M.); premusz.viktoria@pte.hu (V.P.); 3Department of Medical Microbiology, Medical School, Clinical Centre, University of Pécs, H-7624 Pécs, Hungary; 4Institute of Physiotherapy and Sport Sciences, Faculty of Health Sciences, University of Pécs, H-7621 Pécs, Hungary; 5International Research and Innovation in Medicine Program, Cedars-Sinai Medical Center, Los Angeles, CA 90048, USA; sandor.vari@cshs.org

**Keywords:** gestational diabetes mellitus, placenta, gut microbiota, metabolic inflammation, microbial metabolites, inflammation

## Abstract

Gestational diabetes mellitus (GDM) is a multifactorial metabolic disorder arising from impaired insulin sensitivity and altered maternal–fetal energy regulation. Beyond classical mechanisms involving β-cell dysfunction and pregnancy-induced insulin resistance, emerging evidence suggests a bidirectional interaction between the maternal gut microbiota and the placenta, forming a dynamic placenta–gut axis. Microbial dysbiosis alters levels of metabolites, inflammatory mediators, and bile acids, which influence placental signaling, trophoblast metabolism, immune activation, and nutrient transport. Conversely, the placenta secretes hormones, cytokines, lipids, and exosomal miRNAs that shape maternal metabolism and potentially modulate the gut microbiota. This review synthesizes current mechanistic insights underlying the placenta–gut microbiota axis in GDM, describes immune and metabolic crosstalk, and highlights therapeutic opportunities targeting this inter-organ communication system. Addressing these interactions may advance precision strategies for managing GDM and improving outcomes across generations.

## 1. Introduction

Gestational diabetes mellitus (GDM), defined as glucose intolerance first recognized during pregnancy, affects up to 20% of pregnancies globally and is associated with significant short- and long-term metabolic consequences for both mother and offspring [1,2]. Traditionally attributed to an imbalance between pregnancy-induced insulin resistance and β-cell compensatory capacity, GDM is now understood as a complex disorder involving immunological, metabolic, endocrine, and environmental interactions [3].

Two organs have recently emerged as central hubs in GDM pathophysiology: the maternal gut microbiota and the placenta. Pregnancy induces physiological insulin resistance through placental hormones, including human placental lactogen (hPL), placental growth hormone (pGH), progesterone, and estrogens [4]. These hormones activate pathways such as Janus kinase- Signal Transducer and Activator of Transcription (JAK–STAT) and mitogen-activated protein kinase (MAPK), modulating maternal glucose utilization and nutrient partitioning [4]. In GDM, altered hormonal profiles synergize with inflammation and lipotoxicity to impair insulin receptor substrate (IRS) signaling, phosphatidylinositol 3′-kinase (PI3K)/AKT activity, and glucose transporter type (GLUT)4 trafficking in peripheral tissues [5].

Parallel to these endocrine changes, the maternal gut microbiota undergoes profound restructuring across gestation. Normal late pregnancy resembles metabolic states seen in obesity, with reduced microbial diversity and increased pro-inflammatory taxa [6]. In GDM, dysbiosis is more pronounced, featuring enhanced abundance of *Collinsella*, *Ruminococcus*, and *Blautia*, alongside reduced *Akkermansia* and *Bifidobacterium* [7]. Such dysbiosis elevates circulating lipopolysaccharide (LPS), branched-chain amino acids (BCAAs), and microbial-derived bile acids, which activate inflammatory pathways Toll-like Receptor 4 (TLR4)/Nuclear Factor-kappa B (NF-κB) signaling pathway (TLR4/NF-κB), impair insulin signaling, and modulate enterohepatic hormone secretion including glucagon-like peptide 1 and peptide YY [8,9].

The placenta is both a target and mediator of these microbiota-derived signals. Trophoblasts express receptors for microbial metabolites and inflammatory mediators, including G-protein-coupled receptor (GPR)41/43 for short-chain fatty acids (SCFAs), TLR4 for LPS, and Farnesoid X receptor and G protein-coupled bile acid receptor (FXR/TGR5) for bile acids [10]. Activation of these receptors influences key placental processes—nutrient transporter expression (GLUT1, SNATs, FATPs), mitochondrial bioenergetics, angiogenesis, and immune tolerance [11]. GDM placentas show increased oxidative stress, altered mammalian target of rapamycin (mTOR) signaling, and aberrant cytokine and chemokine profiles, all of which can be modulated by gut microbiota-derived factors [12].

Conversely, the placenta impacts the maternal microbiome. Placental hormones drastically change gastrointestinal motility, immunity, and bile acid composition, shaping microbial community structures [13]. Moreover, placenta-derived extracellular vesicles carrying miRNAs, proteins, and lipids circulate systemically and influence maternal tissues, potentially including gut epithelial and immune cells [14,15]. This bidirectional interaction forms a placenta–gut microbiota axis whose disruption contributes to GDM development.

This review synthesizes mechanistic insights into the placenta–gut microbiota axis in GDM, including microbial metabolites, inflammatory and endocrine pathways, exosomal communication, and therapeutic opportunities.

## 2. Gut Microbiota Dysbiosis in Pregnancy and GDM

Pregnancy induces profound physiological and metabolic adaptations that are accompanied by characteristic shifts in gut microbial composition. In normoglycemic pregnancies, the maternal microbiota typically transitions toward a lower-diversity, more pro-inflammatory community structure in late gestation, which is thought to support increased energy harvest and insulin resistance required for adequate fetal growth [13]. One of the earliest demonstrations of gestational shifts in microbiota composition came from Koren et al., who showed that third-trimester stool samples resemble dysbiotic inflammatory states, with expansion of *Proteobacteria* and *Actinobacteria* and reduced richness compared with the first trimester [6]. These compositional changes are considered part of the physiological remodeling of maternal metabolism.

However, in GDM, the extent and character of microbial alterations differ substantially from normal pregnancy (Table 1). Multiple studies describe decreased abundance of beneficial taxa such as *Bifidobacterium* and *Akkermansia*, alongside increased abundance of Gram-negative, LPS-containing bacteria, which may contribute to metabolic endotoxemia and chronic low-grade inflammation [16,17]. Crusell et al. demonstrated that women with GDM show enrichment of *Ruminococcus*, *Parabacteroides*, and *Blautia*, taxa that correlate strongly with glycemic indices and insulin resistance [7]. Moreover, metabolomic analyses suggest that the altered microbiota in GDM may promote elevations in branched-chain amino acids (BCAAs), which are known to impair insulin signaling and predict future diabetes risk [18].

Microbiota-derived metabolites play central mechanistic roles in linking gut dysbiosis to the development of GDM. SCFAs, especially butyrate and propionate, normally serve as anti-inflammatory mediators and regulators of intestinal barrier integrity; reduced SCFA production has been implicated in increased gut permeability, LPS translocation, and systemic inflammation [19,20]. In addition, microbial modulation of bile acid pools affects host metabolism through activation of receptors such as FXR and TGR5, which regulate glucose homeostasis, lipid oxidation, and inflammatory signaling [21]. Dysregulation of these pathways has been observed in metabolic disorders and is thought to contribute to the heightened insulin resistance characteristic of GDM.

Microbiota composition and function also influence maternal immune tolerance, an essential feature of healthy pregnancy. Prince et al. demonstrated that microbial signals regulate maternal mucosal immune responses that shape systemic inflammation and potentially influence materno-fetal immune interactions [13,22]. Perturbations in these pathways may contribute to the exaggerated inflammatory state observed in GDM. Additionally, emerging evidence suggests that altered microbial communities may influence host epigenetic regulation, as maternal metabolic stress and microbial metabolites can modulate chromatin remodeling, histone acetylation, and DNA methylation in tissues relevant to glucose metabolism [23].

**Table 1 ijms-27-00312-t001:** Gut Microbiota Dysbiosis in Healthy Pregnancy versus Gestational Diabetes Mellitus.

Feature	Healthy Pregnancy	Gestational Diabetes Mellitus (GDM)	Mechanistic/Physiological Implications	References
Overall diversity	Reduced diversity in late pregnancy as a physiological adaptation	Further altered diversity; dysbiotic configuration	Exaggerated metabolic inflammation and insulin resistance	[6,24]
Dominant taxa	Enrichment of *Firmicutes* and *Actinobacteria* in late gestation	Increased abundance of *Ruminococcus*, *Blautia*, *Collinsella*, *Parabacteroides*	Correlates with insulin resistance and dysglycemia	[7,25]
Beneficial bacteria	Presence of *Bifidobacterium* and *Akkermansia muciniphila*	Reduced *Bifidobacterium* and *Akkermansia*	Impaired gut barrier integrity and glucose homeostasis	[26,27]
Inflammatory potential	Mild pregnancy-associated inflammation	Increased Gram-negative bacteria and LPS production	Activation of TLR4/NF-κB signaling and systemic inflammation	[26,28]
Metabolic endotoxemia	Low circulating LPS levels	Elevated LPS (“metabolic endotoxemia”)	Impaired insulin signaling and placental inflammation	[26,28]
SCFA production	Adequate SCFA (butyrate, propionate) production	Reduced SCFA availability	Loss of anti-inflammatory signaling and gut barrier dysfunction	[16,29]
BCAA metabolism	Normal microbial regulation of BCAAs	Elevated circulating BCAAs	Insulin resistance and mitochondrial stress	[9]
Bile acid metabolism	Balanced bile acid pools and signaling	Altered bile acid composition and signaling	Dysregulation of FXR/TGR5 pathways affecting glucose metabolism	[16,30]
Immune modulation	Balanced Treg/Th17 immune responses	Shift toward pro-inflammatory immune profile	Exacerbated placental immune activation	[13,31]
Timing of dysbiosis	Gradual changes across gestation	Dysbiosis detectable early in pregnancy	Suggests a predisposing rather than secondary effect	[17,25]

Importantly, longitudinal human cohort studies indicate that microbiome disturbances associated with GDM often precede clinical diagnosis. Fugmann et al. showed that dysbiosis was detectable in early pregnancy among women who later developed GDM, suggesting a causal or predisposing role rather than a consequence of hyperglycemia [7,25]. Fungmann et al. found that diversity metrics and specific taxa abundances during the first trimester predicted glycemic outcomes in the second trimester [25]. These findings highlight the potential of early-pregnancy microbiome signatures as predictive biomarkers for GDM.

Collectively, current evidence strongly supports the concept that gut microbiota dysbiosis contributes to GDM pathogenesis through mechanisms involving inflammation, metabolic endotoxemia, impaired SCFA and bile acid signaling, BCAA accumulation, and immune dysregulation. These disturbances not only influence maternal metabolism but also set the stage for altered placental function and disrupted maternal–fetal nutrient exchange, forming the foundation for the placenta–gut microbiota axis in GDM (Figure 1).

## 3. Molecular Mechanisms Linking Gut Microbiota to Placental Function

Gut microbiota-derived metabolites profoundly influence placental signaling pathways relevant to GDM. SCFAs such as acetate, propionate, and butyrate modulate maternal insulin sensitivity, immune activity, and placental nutrient transport through G-protein-coupled receptors and epigenetic histone modifications, including inhibition of histone deacetylase (HDAC) [16,21]. Reduced and altered SCFA production during dysbiosis contributes to inflammation and impaired trophoblast metabolic flexibility, potentiating placental insulin resistance [26,29]. Dysbiosis also elevates circulating branched-chain amino acids (BCAAs), a metabolic signature tightly linked with insulin resistance and altered mitochondrial function [9]. In addition to the major short-chain fatty acids acetate, propionate, and butyrate, gut microbes also produce minor SCFAs such as valeric acid and isovaleric acid, which are increasingly recognized for their potential signaling roles in host tissues including immune modulation and metabolic regulation. Recent work by Paciolla et al. indicates that these minor SCFAs merit attention as additional gut-derived mediators influencing systemic and placental pathways in metabolic disease contexts [23].

LPS translocation resulting from impaired gut barrier integrity (“leaky gut syndrome”) activates maternal TLR4 signaling and systemic inflammation, extending to the placenta where trophoblasts and macrophages show increased NF-κB activation and cytokine expression [27,28]. Inflammatory cytokines disrupt trophoblast insulin signaling, impair glucose transporter regulation, and alter vascular remodeling, mechanisms that resemble early findings in obesity-related metabolic inflammation [5,26]. The placenta’s nutrient transport capacity—including GLUT1, FATP family members, and system A amino acid transporters—is sensitive to microbial signals, especially in the presence of high LPS or reduced SCFAs, contributing to fetal overgrowth and altered metabolic programming [11,12].

## 4. Immune Crosstalk: Microbiota-Driven Inflammation and Placental Immune Modulation

The maternal immune system represents a central node through which gut microbiota-derived signals influence placental function. Gut dysbiosis promotes differentiation of pro-inflammatory Th17 cells, reduces regulatory T cell (Treg) abundance, and enhances innate immune activation, thereby contributing to chronic low-grade metabolic inflammation characteristic of gestational diabetes mellitus [26,31]. Increased intestinal permeability in dysbiosis facilitates translocation of LPS into the circulation, leading to macrophage activation and elevated production of pro-inflammatory cytokines such as tumor necrosis factor-α (TNF-α) and interleukin-6 (IL-6) [27,28]. These inflammatory mediators reach the placental interface, where they promote trophoblast stress responses and pro-inflammatory polarization of placental macrophages (Hofbauer cells) [17].

Similar immune mechanisms have been described in metabolic disease models, where intestinal microbial shifts directly reshape host immune tone and exacerbate insulin resistance [26,31]. Within the placenta, inflammation amplifies GDM-specific abnormalities, including increased M1 macrophage infiltration, impaired decidual immune tolerance, and heightened inflammasome activation, particularly of the NLRP3 pathway [17]. Cytokine-driven alterations in trophoblast endocrine activity—especially changes in human placental lactogen and placental growth hormone secretion—further reinforce systemic insulin resistance, establishing a feed-forward loop between gut-derived inflammation and placental dysfunction [4,5]. Maternal obesity, which frequently coexists with gut dysbiosis, synergistically intensifies these inflammatory and immune disturbances [26].

Emerging evidence indicates that immune responses at the maternal–fetal interface are not homogeneous across placental compartments, and that distinct immunological patterns in the villous versus extravillous regions are integral to nutrient transport, tolerance, and defense mechanisms. Macrophage polarization also differs markedly between placental compartments. Under physiological conditions, Hofbauer cells within the villous stroma predominantly display an M2-like anti-inflammatory phenotype, supporting tissue remodeling, angiogenesis, and maternal–fetal immune tolerance. In contrast, decidual macrophages located adjacent to extravillous trophoblasts exhibit greater phenotypic plasticity and heightened responsiveness to local chemokine and cytokine gradients. GDM, this balance is disrupted, with multiple studies demonstrating a shift toward an M1-like pro-inflammatory macrophage phenotype, particularly within the extravillous compartment, accompanied by increased expression of TNF-α, IL-6, and IL-1β [32,33].

## 5. Placental Exosomes and miRNAs as Mediators of Microbiota–Placenta Crosstalk

Placental extracellular vesicles (EVs), particularly exosomes, constitute a major communication pathway between the placenta and maternal metabolic tissues. During normal pregnancy, placental exosome release increases progressively; however, gestational diabetes mellitus is associated with a further elevation in circulating placental exosomes enriched in bioactive cargo linked to inflammation, glucose metabolism, and insulin signaling [14,15]. Several placental exosomal microRNAs—including miR-16, miR-21, miR-29, and miR-222—have been implicated in the regulation of insulin sensitivity, adipose inflammation, and endothelial function [14,34].

Importantly, many of these miRNAs are responsive to metabolic and inflammatory cues originating from the gut microbiota. Microbial metabolites such as SCFAs and trimethylamine-N-oxide (TMAO) can modulate miRNA expression through epigenetic mechanisms, thereby influencing trophoblast function, placental barrier integrity, and maternal glucose homeostasis [10,21]. Placental exosomes also exert immunomodulatory effects by regulating macrophage polarization, endothelial activation, and cytokine secretion, further linking microbiota-driven inflammation with systemic metabolic dysregulation [15].

Moreover, EV-mediated transfer of placental miRNAs has been shown to alter insulin signaling pathways in maternal skeletal muscle and liver, reinforcing the contribution of placental exosomes to whole-body metabolic adaptation during pregnancy [14,22]. Emerging evidence suggests that microbial metabolites may influence not only the quantity but also the molecular composition of placental EVs, adding an additional regulatory layer to the placenta–gut microbiota axis [10,22] (Table 2).

## 6. The Placenta Influencing the Maternal Gut Microbiome: Reverse Feedback Mechanisms

Although the impact of the gut microbiota on placental function is well recognized, the reciprocal influence of placental signals on the maternal gut microbial environment has received comparatively less attention. Placental hormones—including progesterone, estrogens, human placental lactogen (hPL), and placental growth hormone—exert profound effects on gastrointestinal motility, mucosal immunity, bile acid metabolism, and nutrient availability, thereby creating selective pressures that shape gut microbial community structure [4,6]. Hormonal modulation of gut physiology is thought to underlie pregnancy-specific shifts in microbiota composition observed even in metabolically healthy pregnancies [6,24].

Placental inflammation characteristic of GDM may further contribute to gut dysbiosis. Elevated circulating cytokines, including IL-6 and TNF-α, as well as markers of oxidative stress, can disrupt intestinal epithelial tight junction integrity and increase gut permeability, thereby promoting an environment favorable to pathogenic bacterial expansion [26,28]. Pregnancy-associated inflammatory signaling has been shown to correlate with alterations in maternal gut microbiota composition and metabolic output, suggesting bidirectional communication between placental immune activation and intestinal microbial ecology [27,31].

In addition to soluble inflammatory mediators, placental EVs may contribute to reverse signaling from the placenta to the maternal gut. Placenta-derived exosomes carrying microRNAs, proteins, and lipids circulate systemically during pregnancy and have been shown to modulate gene expression, immune responses, and metabolic pathways in maternal tissues [14,15]. Although direct evidence for placental exosome uptake by intestinal epithelial cells is still limited, emerging data indicate that EVs can influence epithelial barrier function and immune signaling in distal tissues, raising the possibility that placental exosomes participate in shaping maternal gut microbial colonization patterns [14,22]. This evolving concept highlights a bidirectional placenta–gut communication axis that may amplify metabolic and inflammatory disturbances in GDM.

## 7. Therapeutic and Preventive Interventions Targeting the Placenta–Gut Microbiota Axis

Growing mechanistic insight into the placenta–gut microbiota axis has opened new avenues for preventive and therapeutic strategies in GDM. Interventions aimed at modulating the maternal gut microbiome, improving dietary quality, and targeting downstream molecular pathways hold promise for restoring metabolic and placental homeostasis during pregnancy.

### 7.1. Probiotics and Prebiotics

Multiple clinical and experimental studies indicate that specific probiotic strains, particularly *Lactobacillus* and *Bifidobacterium* species, can improve maternal glycemic control, reduce insulin resistance, and attenuate systemic inflammation during pregnancy [34,35]. These beneficial effects are partly mediated through increased production of SCFAs, enhanced intestinal barrier integrity, and reduced translocation of pro-inflammatory bacterial components such as lipopolysaccharide [16,28]. Improved gut barrier function may, in turn, limit placental exposure to inflammatory mediators and mitigate placental immune activation, processes that are central to GDM pathophysiology.

Prebiotics, including inulin-type fructans and other fermentable fibers, selectively promote the growth of SCFA-producing bacteria and reduce metabolic endotoxemia [36,37]. By lowering circulating LPS levels and dampening inflammatory signaling, prebiotic supplementation may indirectly ameliorate placental inflammation and improve maternal–fetal metabolic exchange.

### 7.2. Dietary Interventions

Dietary patterns exert a profound influence on the composition and metabolic activity of the gut microbiota. High-fiber diets consistently enhance SCFA production, suppress inflammatory pathways, and improve insulin sensitivity in both pregnant and non-pregnant populations [16,31]. Mediterranean-style diets, characterized by high intake of fiber, polyphenols, and unsaturated fatty acids, have been shown to beneficially modulate gut microbiota composition and reduce oxidative stress and inflammation at the placental level [35,37].

Low-glycemic index diets represent another effective nutritional strategy in GDM management. Randomized controlled trials demonstrate that low-glycemic diets improve postprandial glucose control, reduce insulin requirements, and decrease the risk of fetal overgrowth [38]. By attenuating maternal hyperglycemia and insulin resistance, such diets may normalize placental nutrient transporter activity and reduce excessive transplacental glucose and lipid flux, thereby limiting fetal overnutrition.

### 7.3. Pharmacological and Molecular Strategies

Beyond dietary approaches, pharmacological and molecular interventions targeting the placenta–gut axis are gaining increasing attention. Myo-inositol supplementation has demonstrated efficacy in reducing the incidence of GDM and improving insulin sensitivity, particularly in high-risk pregnancies [8,39]. Although its primary mechanism involves insulin signaling, emerging evidence suggests that myo-inositol may also influence gut microbial metabolism and host–microbiome interactions. Precision probiotic and postbiotic therapies designed to target specific inflammatory and metabolic pathways represent a rapidly evolving field. Mechanistic studies highlight the potential of modulating LPS–TLR4 signaling and bile acid receptor pathways, including FXR and TGR5, to improve glucose homeostasis and reduce inflammation [30,40]. Such targeted approaches may offer greater efficacy than conventional probiotics by focusing on defined microbial functions rather than broad taxonomic changes. Finally, advanced drug delivery systems are emerging as innovative therapeutic modalities in pregnancy. EV-based therapies and placenta-targeted nanoparticles are under investigation as means to deliver anti-inflammatory or metabolic modulators directly to placental tissues while minimizing systemic exposure [14,41]. Although still largely experimental, these approaches hold promise for selectively correcting placental dysfunction linked to gut microbiota-derived inflammatory signals.

## 8. Vaginal Microbiome and Vertical Transmission in Gestational Diabetes Mellitus

The maternal vaginal microbiome represents a crucial microbial niche that undergoes dynamic remodeling during pregnancy and serves as one of the earliest microbial exposures for the neonate during vaginal birth. In healthy pregnancies, the vaginal microbiota is typically characterized by *Lactobacillus* dominance, low diversity, and an acidic environment that protects against pathogenic colonization and inflammation [42]. *Lactobacillus* species—including *L. crispatus*, *L. jensenii*, and *L. iners*—produce lactic acid, hydrogen peroxide, and antimicrobial peptides that maintain mucosal immune homeostasis and contribute to neonatal microbial seeding during delivery [43]. Emerging evidence indicates that gestational diabetes mellitus (GDM) is associated with significant alterations to the vaginal microbiome. Studies show that women with GDM exhibit a higher vaginal microbial diversity, reduced abundance of protective *Lactobacillus* species, and increased colonization by opportunistic bacteria such as *Gardnerella*, *Atopobium*, and *Ureaplasma* [44]. This shift toward a more diverse, less *Lactobacillus*-dominant community is reminiscent of community state types associated with bacterial vaginosis and correlates with heightened local inflammation, altered epithelial barrier function, and increased susceptibility to ascending infections [45]. Hyperglycemia may drive these shifts by altering mucosal immunity, increasing glycogen availability, and modifying vaginal epithelial metabolic activity [6] (Table 3).

### 8.1. Vertical Transmission and Neonatal Microbiome Programming

The vaginal microbiome plays a central role in vertical microbial transmission during vaginal delivery. Infants delivered vaginally acquire maternal vaginal and fecal microbes, which seed the neonatal gut and influence early immune development, metabolic programming, and intestinal barrier maturation [56]. Dysbiosis in the maternal vaginal microbiome—such as the *Lactobacillus*-depleted communities observed in GDM—has been linked to altered initial microbial colonization patterns in newborns, including increased abundance of *Streptococcus*, *Staphylococcus*, and *Enterobacteriaceae* [57]. These microbial signatures are associated with higher risk of early-life inflammation, impaired glucose homeostasis, and increased susceptibility to metabolic disorders later in childhood. Several studies demonstrate that infants born to GDM mothers display distinct gut microbiota profiles as early as 3–6 months, independent of delivery mode, suggesting that vertical transmission may occur not only during birth but also via transplacental microbial metabolites, breastmilk microbiota, and maternal skin contact [7]. Elevated maternal blood glucose and associated inflammation may modify the pool of metabolites and immune factors transferred to the fetus, further influencing neonatal microbial colonization and metabolic programming (Figure 2).

### 8.2. Mechanistic Links Between Vaginal Dysbiosis, Placental Immunity, and GDM

Beyond neonatal effects, vaginal microbiota alterations may impact pregnancy outcomes by interacting with the maternal immune system and the placenta. Increased abundance of anaerobic taxa such as Gardnerella and Prevotella is associated with elevated pro-inflammatory cytokines (IL-6, TNF-α) and chemokines in the lower reproductive tract, which may propagate upward to the decidua and placenta [58]. Such inflammatory signaling can exacerbate trophoblast stress, impair nutrient transport, and contribute to placental metabolic dysfunction—pathways already dysregulated in GDM. Moreover, vaginal microbes or their components may translocate in rare cases, influencing intrauterine inflammation and potentially modulating placental epigenetic programming.

### 8.3. Delivery Mode as a Modulator of Microbial Inheritance in GDM

The interaction between GDM, delivery mode, and neonatal microbiome development is complex. Cesarean delivery bypasses vaginal microbial transfer, leading to delayed colonization by beneficial taxa and increased prevalence of skin-associated microbes such as *Staphylococcus* and *Corynebacterium* [59]. Because women with GDM have higher rates of cesarean delivery, this further compounds the risk of altered neonatal microbiome maturation. Some researchers propose postnatal interventions such as maternal vaginal microbial “seeding” or targeted probiotic supplementation, although evidence remains preliminary and controversial.

### 8.4. Implications for Prevention and Intervention

Understanding the role of the vaginal microbiome in GDM adds an important dimension to maternal–fetal microbial crosstalk. Interventions aimed at restoring *Lactobacillus* dominance—such as probiotic *Lactobacillus* therapy, glycemic control optimization, or vaginal microbiome–directed treatments—may have potential to improve both maternal vaginal health and neonatal microbiome programming. Longitudinal studies integrating vaginal, gut, and placental microbiome profiling are required to fully elucidate the role of vaginal dysbiosis in GDM progression and offspring metabolic risk.

## 9. Future Directions

Despite progress, major gaps remain in understanding the placenta–gut microbiota axis. Precise mapping of which microbial metabolites reach the placenta in physiologically relevant concentrations is lacking. Longitudinal studies bridging early pregnancy microbiome profiles with later placental transcriptomic and exosome signatures are needed. Moreover, interindividual heterogeneity—including fetal sex, maternal genetics, and dietary patterns—modifies host–microbiome interactions and is insufficiently incorporated into current models [60,61]. Emerging multi-omics integration promises deeper mechanistic insights, but harmonized protocols and larger cohorts will be essential. Ultimately, leveraging microbiota–placenta communication for targeted preventive and therapeutic strategies could transform GDM management.

## 10. Conclusions

Gestational diabetes mellitus arises from a complex interplay between maternal metabolic adaptations, inflammatory signals, and hormonal changes during pregnancy. Increasing evidence now positions the placenta–gut microbiota axis as a central regulatory network in this disorder. Gut dysbiosis contributes to metabolic endotoxemia, altered microbial metabolite production, and immune activation, all of which influence placental signaling, nutrient transport, and trophoblast function. In parallel, the GDM placenta releases hormones, cytokines, and extracellular vesicles that can reshape the maternal gut environment, indicating that communication between these two systems is bidirectional rather than unidirectional (Figure 3).

This reciprocal relationship creates a feed-forward loop in which microbiota-derived molecules (e.g., LPS, SCFAs, bile acids) modulate placental metabolism and immune responses, while placental stress signals (e.g., IL-6; TNF-α, miRNAs) further destabilize gut barrier integrity and microbial composition. Understanding this systems-level crosstalk provides new mechanistic insights into why some women develop GDM despite similar metabolic or genetic backgrounds.

Therapeutic approaches targeting the placenta–gut axis—including precision probiotics, postbiotics, dietary modulation, micronutrients, and exosome-based strategies—represent promising avenues to restore metabolic homeostasis. However, most interventions remain in early stages, and mechanistic data connecting microbiome modulation to placental improvement are still limited. Future work should integrate multi-omics profiling, longitudinal sampling, mechanistic animal models, and human intervention trials to unravel causal pathways and identify biomarkers predictive of disease onset or treatment response.

Ultimately, recognizing GDM as a disorder of inter-organ communication rather than isolated maternal hyperglycemia reframes our scientific and clinical approach. Targeting the microbiota–placenta interface offers the potential not only to improve pregnancy outcomes but also to reduce long-term metabolic risks for both mother and child, making it a highly compelling frontier in maternal–fetal medicine.

## Figures and Tables

**Figure 1 ijms-27-00312-f001:**
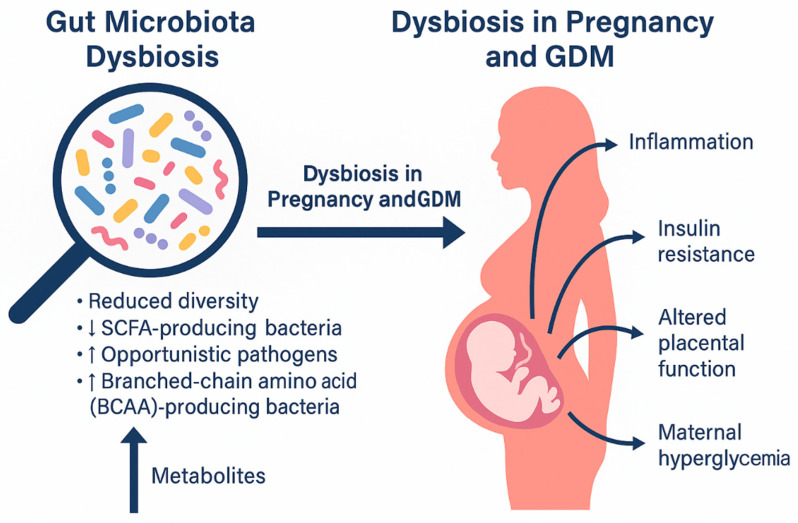
Dysbiosis in pregnancy and gestational diabetes mellitus.

**Figure 2 ijms-27-00312-f002:**
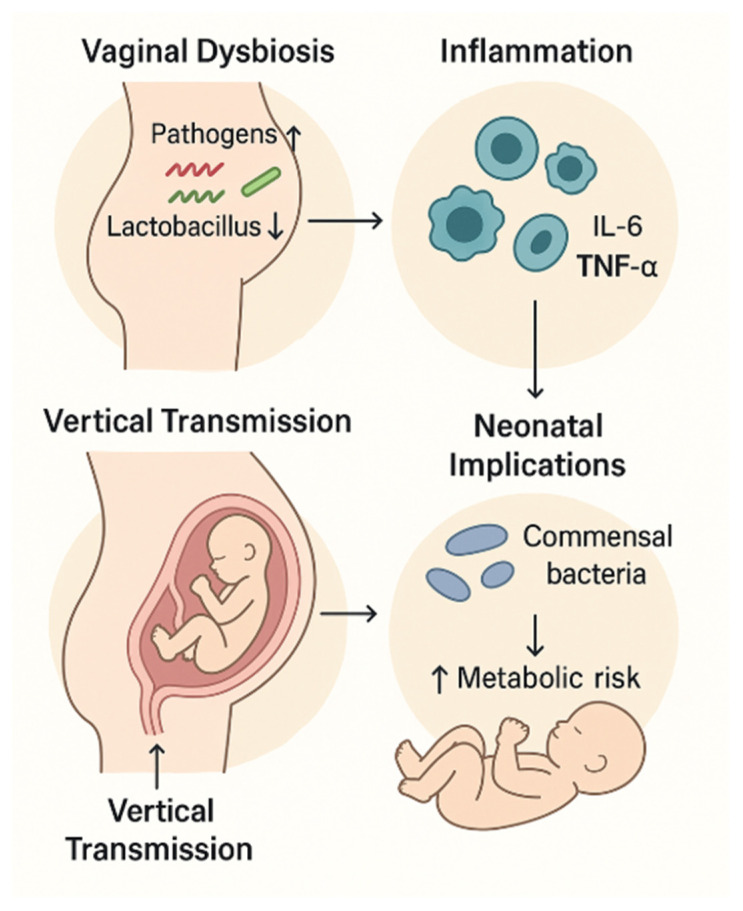
Vaginal microbiome and vertical transmission.

**Figure 3 ijms-27-00312-f003:**
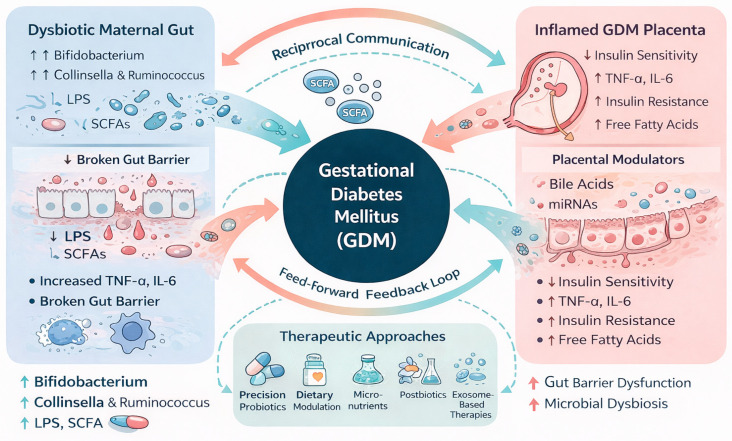
Placenta–gut microbiota axis.

**Table 2 ijms-27-00312-t002:** Placental Exosomes and miRNAs Involved in Microbiota–Placenta Crosstalk in Gestational Diabetes Mellitus.

Component	Source	Key Molecular Cargo	Target Tissue/Cells	Biological Effects Relevant to GDM	References
Placental exosomes (EVs)	Syncytiotrophoblasts; increased release in GDM	miRNAs, proteins, lipids	Maternal liver, adipose tissue, skeletal muscle	Modulation of insulin signaling, glucose uptake, and inflammatory pathways	[14,15]
miR-16	Placental exosomes; upregulated by metabolic stress	miR-16	Endothelial cells, adipocytes	Regulation of inflammation and insulin signaling; endothelial dysfunction	[14,22]
miR-21	Placental exosomes; influenced by inflammatory signals	miR-21	Immune cells, insulin-responsive tissues	Promotes inflammation and insulin resistance via NF-κB and MAPK signaling	[17,26]
miR-29 family (miR-29a/b/c)	Placental exosomes; responsive to SCFA and metabolic stress	miR-29	Liver, skeletal muscle	Impaired insulin signaling and glucose metabolism; epigenetic regulation	[9,20]
miR-222	Placental exosomes; elevated in GDM	miR-222	Adipose tissue, muscle	Disruption of insulin receptor signaling and adipokine regulation	[14,22]
SCFA-modulated exosomal cargo	Gut microbiota-derived SCFAs (butyrate, propionate)	Altered miRNA and protein profiles	Placenta, maternal metabolic tissues	Anti-inflammatory effects; improved insulin sensitivity; epigenetic modulation via HDAC inhibition	[16,21]
LPS-driven EV alterations	Gut dysbiosis; increased intestinal permeability	Pro-inflammatory miRNAs, cytokine-associated proteins	Placenta, immune cells	Enhanced placental inflammation, macrophage activation, insulin resistance	[27,28]
EV-mediated immune modulation	Placental stress and inflammation	miRNAs regulating macrophage polarization	Maternal immune cells	Shift toward pro-inflammatory (M1) phenotype; impaired immune tolerance	[15,31]
Potential biomarker EVs	Early pregnancy placental exosome release	Circulating miRNA signatures	Maternal plasma	Early prediction of GDM risk and disease severity	[15,20]

**Table 3 ijms-27-00312-t003:** Differences in the Vaginal Microbiome in GDM vs. Healthy Pregnancy.

Feature	Healthy Pregnancy	GDM Pregnancy	Physiological Implication	References
Dominant taxa	Lactobacillus-dominated microbiome (mainly *L. crispatus*, *L. jensenii*, *L. gasseri*)	Reduced *Lactobacillus* spp.; increased *Gardnerella*, *Atopobium*, *Prevotella*, *Ureaplasma*	Elevated vaginal pH, weakened epithelial defense, increased susceptibility to inflammation and infection	[39,46]
Microbial diversity	Low diversity, stable community	Increased alpha-diversity with enrichment of anaerobic taxa	Shift toward dysbiotic profiles resembling bacterial vaginosis	[39,47,48]
Community State Types (CSTs)	CST I–III (*Lactobacillus*-dominant)	Shift toward CST IV (diverse anaerobes)	Increased inflammatory signaling and mucosal barrier disruption	[47,48,49]
pH regulation	Low pH (≈3.5–4.5) maintained by lactic acid production	Elevated pH due to reduced lactic acid production	Loss of antimicrobial protection and altered immune tone	[42,50]
Inflammatory milieu	Physiological low-grade inflammation of pregnancy	Increased vaginal IL-6, IL-8, TNF-α	Local inflammation may propagate to decidua and placenta	[21,51]
Metabolic environment	Normal glycogen metabolism supporting *Lactobacillus* growth	Altered glycogen and lactate metabolism	Favors anaerobic overgrowth and dysbiosis	[50,52]
Vertical transmission potential	Predominant transfer of beneficial *Lactobacillus* to neonate during vaginal delivery	Increased likelihood of dysbiotic microbial transfer	Altered early-life microbiome seeding	[53,54]
Neonatal implications	Healthy gut colonization and immune maturation	Altered infant gut microbiota; increased metabolic risk	Early life metabolic and immune programming effects	[53,54,55]

## Data Availability

No new data were created or analyzed in this study. Data sharing is not applicable to this article.

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
