# Peer review of "The Placenta–Gut Microbiota Axis in Gestational Diabetes Mellitus: Molecular Mechanisms, Crosstalk, and Therapeutic Perspectives"

_ijms, 2025, doi:10.3390/ijms27010312_

Round 1
Reviewer 1 Report
Comments and Suggestions for Authors
The manuscript titled "The Placenta–Gut Microbiota Axis in Gestational Diabetes 2 Mellitus: Molecular Mechanisms, Crosstalk, and Therapeutic Perspectives" explores a relevant and timely topic.
The text is well-developed and presents a coherent biological rationale.
The review findings are presented comprehensively, and the figures effectively represent trajectory patterns, although their quality could be improved; the smaller fonts are difficult to read. I suggest saving and submitting as a JPEG or PNG for better quality and, if possible, using a different font.
Overall, the text is useful for contextualizing the findings within the existing literature. However, I suggest expanding the "immune crosstalk and placental immune modulation" section to also address specific immunological differences in placental compartments, since recently published studies have shown that cytokine profiles and macrophage polarization patterns differ between villous and extravillous placental compartments. Adding a note on these differences would strengthen the interpretation of placental immune regulation and better reflect the current understanding of the specific immunological heterogeneity of each compartment at the maternal-fetal interface.
I recommend a careful review of the scientific language throughout the manuscript, paying particular attention to the use of italic for scientific names when necessary and other formally defined terms.
I also suggest updating the information in Reference 1 of the American Diabetes Association. The most recent version of the ADA Standards of Care (2026) has already been published online.
Overall, the manuscript is well written and scientifically sound. Furthermore, English is excellent.
Author Response
Dear Reviewer,
We would like to thank the careful evaluation of our manuscript entitled “The Placenta–Gut Microbiota Axis in Gestational Diabetes Mellitus: Molecular Mechanisms, Crosstalk, and Therapeutic Perspectives.” We greatly appreciate the constructive comments and valuable suggestions, which have helped us to improve the clarity, depth, and overall quality of the manuscript.
We have addressed all comments point by point below. All changes have been incorporated into the revised manuscript and are highlighted accordingly.
The manuscript titled "The Placenta–Gut Microbiota Axis in Gestational Diabetes 2 Mellitus: Molecular Mechanisms, Crosstalk, and Therapeutic Perspectives" explores a relevant and timely topic.
The text is well-developed and presents a coherent biological rationale.
- The review findings are presented comprehensively, and the figures effectively represent trajectory patterns, although their quality could be improved; the smaller fonts are difficult to read. I suggest saving and submitting as a JPEG or PNG for better quality and, if possible, using a different font.
We have corrected the fonts.
- Overall, the text is useful for contextualizing the findings within the existing literature.
Thank you.
- However, I suggest expanding the "immune crosstalk and placental immune modulation" section to also address specific immunological differences in placental compartments, since recently published studies have shown that cytokine profiles and macrophage polarization patterns differ between villous and extravillous placental compartments. Adding a note on these differences would strengthen the interpretation of placental immune regulation and better reflect the current understanding of the specific immunological heterogeneity of each compartment at the maternal-fetal interface.
We thank the reviewer for this insightful suggestion. We have expanded the “Immune Crosstalk and Placental Immune Modulation” section to explicitly address immunological heterogeneity across placental compartments. The revised text now highlights differences between villous and extravillous placental regions, including compartment-specific cytokine profiles and macrophage polarization patterns, and discusses how these distinctions may influence microbiota-driven inflammatory signaling in GDM. Relevant studies have been added to support this expanded interpretation.
- I recommend a careful review of the scientific language throughout the manuscript, paying particular attention to the use of italic for scientific names when necessary and other formally defined terms.
Thank you for this comment. We have revised the manuscript accordingly.
- I also suggest updating the information in Reference 1 of the American Diabetes Association. The most recent version of the ADA Standards of Care (2026) has already been published online.
Overall, the manuscript is well written and scientifically sound. Furthermore, English is excellent.
Thank you, we have changed Ref.1.
Sincerely,
Reka Vass MD PhD
Reviewer 2 Report
Comments and Suggestions for Authors
The manuscript is valuable. The manuscript provides a comprehensive and timely review of the "Placenta–Gut Microbiota Axis" in the context of Gestational Diabetes Mellitus (GDM). The authors move beyond the classical "gut-centric" view to propose a bidirectional model where the placenta acts as both a dsensor of microbial signals and an active modulator of the maternal microbiome. The inclusion of the vaginal microbiome and its role in neonatal programming adds also a significant clinical value. The figures and tables are well-structured and greatly help in synthesizing complex molecular pathways.
The only raccomandation is to include also the mention of minor SCFA.
Among the SCFAs, minor metabolites have to be also considered due to their biological function, also in light of GPR41/43 receptors (refer to DOI: 10.3390/cells14221823). This point is important because of the involvment epigenetic regulation of HDAC, crossing the molecular pathways the authors reported in their manuscript, sistemic inflammation and central nervous system homeostasis.
Author Response
Dear Reviewer,
We would like to thank the careful evaluation of our manuscript entitled “The Placenta–Gut Microbiota Axis in Gestational Diabetes Mellitus: Molecular Mechanisms, Crosstalk, and Therapeutic Perspectives.” We greatly appreciate the constructive comments and valuable suggestions, which have helped us to improve the clarity, depth, and overall quality of the manuscript.
We have addressed all comments point by point below. All changes have been incorporated into the revised manuscript and are highlighted accordingly.
The manuscript is valuable. The manuscript provides a comprehensive and timely review of the "Placenta–Gut Microbiota Axis" in the context of Gestational Diabetes Mellitus (GDM). The authors move beyond the classical "gut-centric" view to propose a bidirectional model where the placenta acts as both a dsensor of microbial signals and an active modulator of the maternal microbiome. The inclusion of the vaginal microbiome and its role in neonatal programming adds also a significant clinical value. The figures and tables are well-structured and greatly help in synthesizing complex molecular pathways.
Thank you.
The only raccomandation is to include also the mention of minor SCFA.
Among the SCFAs, minor metabolites have to be also considered due to their biological function, also in light of GPR41/43 receptors (refer to DOI: 10.3390/cells14221823). This point is important because of the involvment epigenetic regulation of HDAC, crossing the molecular pathways the authors reported in their manuscript, sistemic inflammation and central nervous system homeostasis.
Thank you for this valuable suggestion. We have expanded the discussion of microbial metabolites to include minor short-chain fatty acids, such as valeric acid, and their potential signaling roles in metabolic and inflammatory pathways. This has been added to Section 3.
Sincerely,
Reka Vass MD PhD